# PHI-FORMER: A PAIRWISE HIERARCHICAL APPROACH FOR COMPOUND-PROTEIN INTERACTIONS PREDICTION

## ABSTRACT

Predicting compound-protein interactions (CPIs) is critical for AI-aided design. Recent deep learning (DL) methods have successfully modeled molecular interactions at the atomic level, achieving both efficiency and accuracy improvements compared to traditional energy-based methods. However, these models do not always align with the chemical realities of CPIs, as molecular fragments (i.e., motifs) often participate in the interactions dominantly. In this paper, we propose a pair-wise hierarchical interaction representation learning (Phi-former) method to fill this gap by considering the role of motifs for CPIs. Phi-former represents the compound or protein hierarchically and employs a pair-wise specific pre-training framework to model the interactions in a more systematic way (i.e., atom-atom, motif-motif, and atom-motif). We propose an intra-level and inter-level Phi-former pipeline for learning the pair-wise biomolecular graph representation, making learning the different interaction levels mutually beneficial. We demonstrate that Phi-former can achieve superior performance on CPI-related tasks. Furthermore, a case study indicates that our method can accurately identify the specific atoms or motifs activated in CPIs, and thus provide good model explanations that may give insights into molecular structural optimization.

## 1 INTRODUCTION

Learning compound-protein interaction (CPI) is an essential task in drug discovery, which involves identifying and characterizing the molecular interactions between small organic molecules (compounds) and their target proteins. Interactions between small molecules and proteins are also pivotal in many biological processes. However, experimentally characterizing these interactions is time-consuming (Jorgensen & Thomas, 2008), labor-intensive, and expensive, highlighting the need for efficient and accurate computational methods for tackling CPI-related tasks.

To better capture the interactions in three-dimensional (3D) space, recent deep learning (DL) (Goodfellow et al., 2016) methods (Stärk et al., 2022; Lu et al., 2022; Zhang et al., 2022; Huang et al., 2022) represent the compound-protein pair with graph-structured data. They tend to represent molecular atoms with graph nodes and physical relationships between atoms with edges. In this way, CPI-related tasks such as binding affinity prediction are addressed by modeling all interactions at the atom-atom level. Despite the promising performance, these methods ignore a critical chemical principle: molecular motifs typically function as a whole in CPIs. Without modeling motif-activated interactions, the learned atom-atom level interactions may not include complete functional groups (motifs), leading to biased predictions for downstream CPI tasks. For instance, in Figure 1, our objective is to predict the binding conformation of the protein (purple) and the compound (green). Figure 1C represents the predicted conformation, while Figure 1B serves as the ground truth. Examining this interaction at the atomic level and considering fundamental chemistry principles, hydrogen bonds typically do not prefer forming between carbon (C) and nitrogen (N) atoms. Consequently, neglecting the consideration of molecular motifs may result in an inaccurate predicted binding pose, as demonstrated in Figure 1C, where a hydrogen bond is erroneously formed between the C and oxygen (O) atoms. As evident from the ground truth, the interaction between these atoms is not driven solely by individual atomic propensities. Instead, it is facilitated by the affinity of the carbonyl and pyridine functional groups to establish a hydrogen bond between the C and N atoms.

There are studies focused on hierarchical structure modeling for biomolecules (Zhang et al., 2021; Zang et al., 2023). However, these models are specifically designed for single-molecule graphs, rendering them unsuitable for direct application to CPI tasks. Furthermore, the lack of constraints among different levels can lead to biased predictions in CPI tasks. As illustrated in Figure 1B, the presence of constraints imposed by the carbonyl and pyridine functional groups facilitates the formation of a weak interaction between the C and N atoms. Without these constraints, an erroneous weak interaction is formed between the C and O atoms at the atomic level.

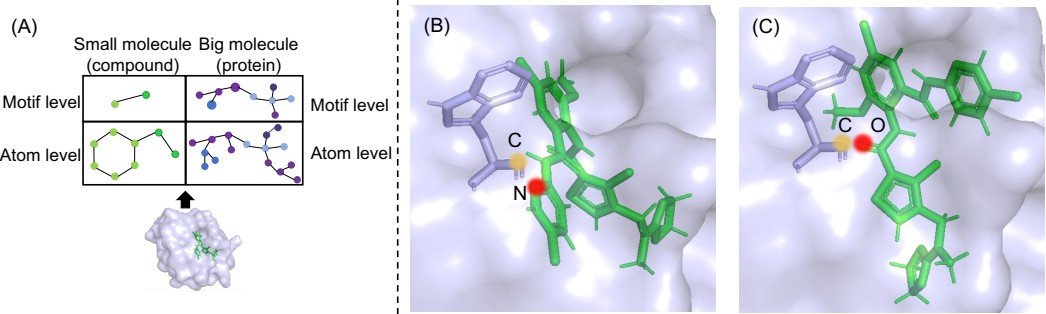

Figure 1: (A) The hierarchical levels of protein and compound. The purple surface represents the protein, while the green ligand denotes the compound. (B) The purple ball-and-stick model illustrates a residue within the protein, showcasing the weak interaction between yellow C and red N atoms. (C) An erroneous binding result of the weak interaction between the yellow C and red O atoms.

To tackle the aforementioned challenges, we have devised a novel transformer-based **p**air-wise **h**ierarchical **i**nteraction representation learning framework, referred to as **Phi**-former. This framework employs a pre-training and fine-tuning structure and is specifically designed to account for interactions at both atomic and motif levels. For the hierarchical model component, the inter and intra losses enable the learning of different interaction levels to be mutually beneficial. The model is in accordance with the chemical facts by conditioning the atom-atom interaction on the motif-motif interaction.

We design three representative CPI tasks in the fine-tuning phase: binding affinity prediction (Wan & Zeng, 2016), drug-target interaction (DTI) (Bleakley & Yamanishi, 2009), and docking pose generation. Our pre-trained model demonstrates promising performance across these tasks. Additionally, we conduct case studies to showcase the ability of our pre-training model to accurately capture various interaction types between different levels (i.e., atom-atom, motif-motif, and atom-motif).

To summarize, our model offers the following contributions:

- We employ a hierarchical graph and a pretraining framework to model the interaction between compound and protein.

- We design inter and intra losses on pre-training tasks, facilitating the mutual benefit of learning across different interaction levels.

- We arrange downstream tasks to demonstrate the effectiveness and robustness of our model, and we conduct case studies to justify that our model is self-consistent on chemical rules.

## 2 RELATED WORKS

**Graph Encoder.** Biomolecules naturally possess graph-based topological structures, so much research involves using graph models to represent biomolecules. The most common graph encoders are **Graph Neural Networks (GNNs)**, such as GCN (Kipf & Welling, 2016), GraphSAGE (Hamilton et al., 2017), MPNN (Gilmer et al., 2017), and GIN (Xu et al., 2018). With the rise of using graph models for biomolecules, **graph transformer** models that were previously too large for traditional graphs have gained prominence, such as Graphormer (Ying et al., 2021), Transformer-M (Luo et al.,

2022), Unimol (Zhou et al., 2023), and GEM-2 (Liu et al., 2022). These graph encoders perform exceptionally well on various tasks, like molecular property prediction, binding affinity prediction, drug-target interaction, and binding pose prediction.

**Hierarchical modeling on biomoleculars** The models under discussion predominantly focus on the representation of atomic level graphs. Recent research by (Yu & Gao, 2022) and (Zang et al., 2023) postulate that both the atomic level structure and the secondary structure of molecules play a crucial role in enhancing the performance of downstream tasks involving individual biomolecules. This viewpoint, referred to as the **hierarchical molecular graph model**, has been further extended to Compound-Protein Interaction (CPI) tasks by (Dou et al., 2023) and (Bui-Thi et al., 2022). Nonetheless, these investigations primarily centered on the hierarchical structure within a single biomolecular graph and did not explicitly model the interactions.

**Binding Affinity Prediction.** Binding affinity in compound-protein interaction (CPI) tasks quantifies the potency of intermolecular associations between chemical compounds and proteins. Given a protein sequence and a Simplified Molecular Input Line Entry System (SMILES) representation (Weininger, 1988) of the compound, various models such as MONN (Li et al., 2020) and TankBind (Lu et al., 2022) are capable of predicting binding affinity without requiring knowledge of the binding conformation. In contrast, models such as OnionNet (Zheng et al., 2019), IGN (Jiang et al., 2021), SIGN (Li et al., 2021), SS-GNN (Zhang et al., 2022), GraphDTA (Nguyen et al., 2021), and Transformer-M (Luo et al., 2022) necessitate 3D information to enhance their modeling capabilities, thus relying on the binding conformation of a compound and protein as input. Specifically, Transformer-M operates on a singular molecular graph, and its pre-training on molecular properties enables it to achieve promising performance in binding affinity prediction tasks.

# 3 METHOD

To address the above-mentioned problems, we propose a framework containing the pre-training model based on hierarchy graphs and the fine-tuning model, aiming to learn the interactions in different levels of biomolecules and achieve better performance in downstream tasks. Figure 2 shows the framework of our model. The framework includes the pre-training part and fine-tuning part.

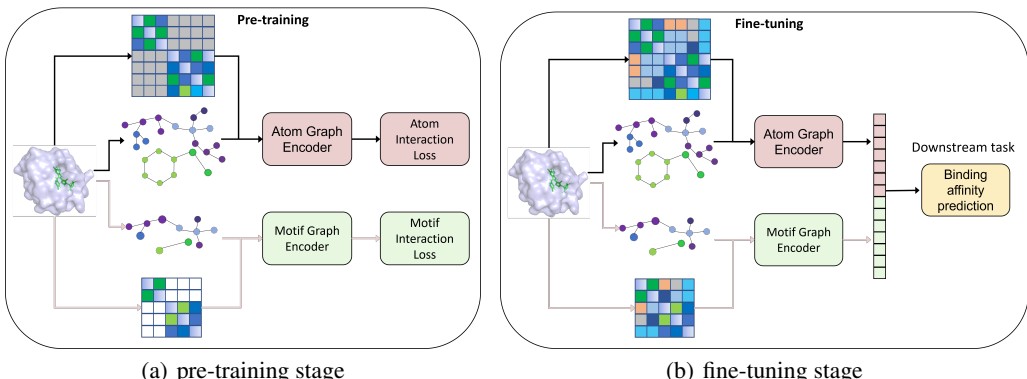

(a) pre-training stage        (b) fine-tuning stage

Figure 2: (a) the pre-training process. In this phase, a complex structure is represented using atom graphs and motif graphs, while the distances between the nodes of the compound and protein are manually masked. (b) the fine-tuning process, during which complete information is provided for some downstream task. The output representations of the atom and motif graphs are then utilized to generate the final prediction.

## 3.1 MOTIF GRAPH

We initially establish a motif level graph to represent the hierarchical architecture of the graph. Given an atomic level graph $G(V, E, P)$, where $V = \{v_1, v_2, \ldots, v_n\}$ symbolizes the atoms, $E =$

$$\begin{cases} \text{single bond} & \text{break} \\ \text{double bond\&triple bond} & \text{keep} \\ \text{bond in a ring} & \text{keep} \end{cases}$$

Figure 3: motif graph generation rule

$\{e_1, e_2, \ldots, e_n\}$ denotes the chemical bonds, and $P = \{p_1, p_2, \ldots, p_n\}$ signifies the Euclidean coordinates.

We delineate a motif graph $\mathrm{T}(M, E', Q)$ and a function $\Theta(*)$ from a graph G to a motif graph T:

$$\Theta(G) : G(V, E, P) -> \mathrm{T}(M, E', Q) \tag{1}$$

Here, $M = \{m_1, m_2, \ldots, m_n\}$, $Q = \{q_1, q_2, \ldots, q_n\}$, $q_1 = avg\{q_1, q_2, \ldots, q_k\}$, $m_1 = \{v_1, v_2, \ldots, v_k\}$, and $E' \subseteq E$.

For the function $\Theta(*)$, we dissociate every torsional bond in molecular structures to obtain the motif graph vertices as shown in Figure 3.1, It can be observed that the single torsional bond is dissociated, while the double bond, triple bond, and bonds in rings are retained.

Specifically for proteins, we retain the polypeptide backbone in every amino acid residue, as shown in Figure 1A. This signifies that we solely cleave the sidechains because the interaction with the compound typically transpires at the sidechains rather than the polypeptide backbone.

Upon constructing the motif level graph, the centroid point's location is employed to represent the motif's position, and the mean initial embedding of the atom within the motif serves as the initial embedding of the motif.

## 3.2 GRAPH TRANSFORMER

Considering both the atomic-level graph and the motif level graph, we employ encoders to embed these graphs into a latent space, thereby obtaining their respective representations. Appropriate models for handling data exhibiting graph structures encompass Graph Neural Networks (GNNs) (Kipf & Welling, 2016) and graph transformers (Ying et al., 2021). In this study, we opt for graph transformers for two primary reasons. First, GNN models typically exhibit shallow architectures due to the over-smoothing issue, rendering them incapable of capturing long-range information. Second, our objective is to utilize a pre-training and fine-tuning scheme to more effectively model hierarchical interactions. Within this scheme, GNN parameters are relatively small, making it challenging to circumvent overfitting. Consequently, we incorporate graph transformers in our proposed models. Figure 4 shows the architecture of our graph transformer encoder.

### 3.2.1 SPATIAL POSITIONAL ENCODING (SPE)

Positional Encoding for Transformers in NLP Tasks (Ke et al., 2020) is designed to differentiate words based on their positions within a sentence. In the context of graphs with 3D spatial information, it is essential to distinguish nodes according to their distinct coordinates. Consequently, we introduce a spatial positional encoding method. To ensure the $E(n)$ invariance of the model, i.e., maintaining consistent prediction results when the input graph undergoes rotation or translation, we utilize Gaussian kernels to encode the Euclidean distance into a spatial positional encoding matrix. This approach maintains the relative distance, which remains unchanged during rotation and translation. Given a pair of Euclidean distances $d_{ij}$ $(Dis(v_i, v_j))$, we compute the spatial positional encoding by

$$s_{ij}^k = \{\mathcal{G}(\alpha t_{ij} d_{ij} + \beta t_{ij}), \mu^k, \sigma^k) \mid k \in [1, C]\} \tag{2}$$

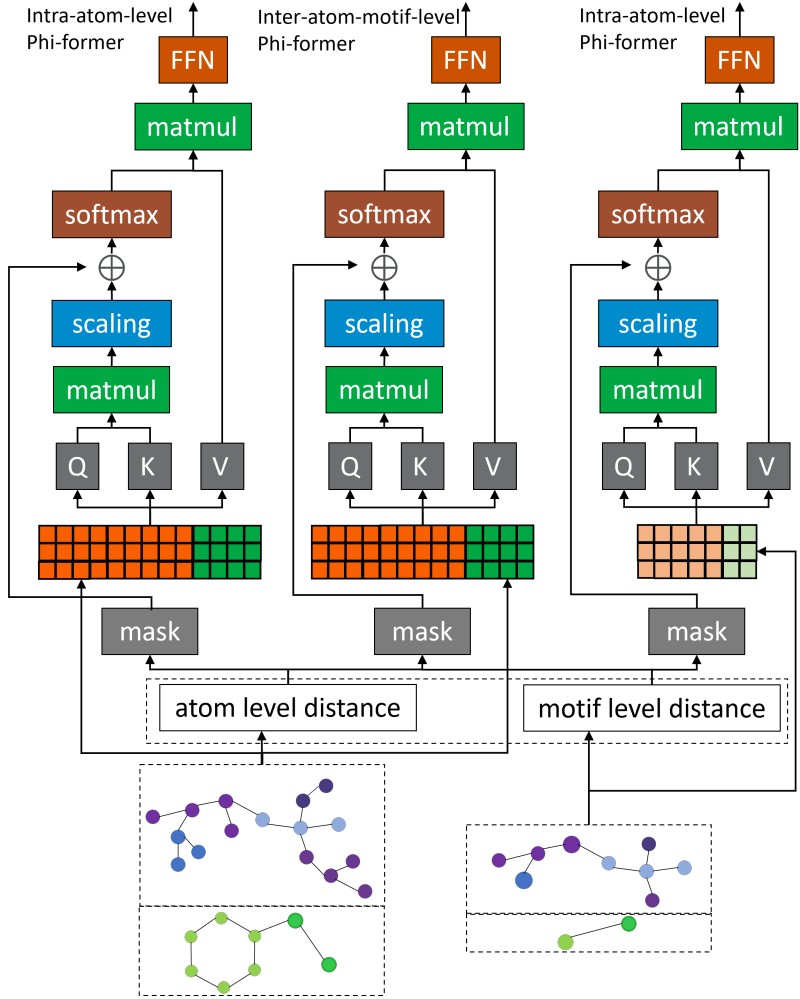

Figure 4: Graph transformer architecture as encoder

let $G$ represent the Gaussian basis functions, with $\mathcal{G}(d, \mu, \sigma) = \frac{1}{\sigma\sqrt{2\pi}}e^{-\frac{(d-\mu)^2}{2\sigma^2}}$. The edge type $t_{ij}$ between nodes $i$ and $j$ is determined by their respective node types. The parameters $\alpha$, $\beta$, $\mu^k$, and $\sigma^k$ are all trainable. Lastly, $C$ denotes the desired number of output channels, expressed as $n * n * C$.

Here we introduce a mapping function $\phi(G)$ that transforms an input graph into an initial embedding matrix $X$ and a spatial positional encoding $S$. This can be represented as:

$$X, S = \phi(G) \tag{3}$$

Note that the input graph $G$ can be either an atomic-level graph or a motif-based graph. The particular methodology employed for input representation generation is delineated in Appendix A.1.

### 3.2.2 ATTENTION BIAS IN DIFFERENT STAGE

Considering a spatial positional encoding $S$, our objective is to employ it for discerning distinct node locations in the spatial domain. Consequently, we incorporate the spatial positional encoding as an additional component to the existing encoding scheme

$$attn(X) = \text{softmax}\left(\frac{XW_q\left(XW_k\right)^\top}{\sqrt{d}} + S\right) \tag{4}$$

where $X$ is the input node embedding matrix, $W_q$, $W_k$ are trainable matrices.

Given an input $X$ consisting of $m$ compound nodes and $n$ protein nodes. The upper-left segment corresponds to the compound-specific spatial positional encoding $S_c$, while the lower-right segment represents the protein-specific spatial positional encoding $S_p$. The remaining matrix elements are populated with either 0 or alternative initial values, which will be elaborated upon in Section 3.3.

In the subsequent phase of the downstream task, given that we possess the complete coordinates of the complex, we can efficiently populate the upper-right and lower-left sections with the cross-distance SPE values, specifically $S_{cp}$ and $S_{pc}$.

For the motif level graph, the centroid location is employed to depict the motif's position, thereby utilizing an identical approach for computing the SPE as employed in the atomic level graph.

With an initial embedding matrix $X$ and a spatial positional encoding $S$ defined, we can define a graph transformer, denoted as $GE$. $GE$ can be expressed as

$$H = GE(X, S) \tag{5}$$

Here, $H$ represents a combined set of node representations, encompassing both compound and protein nodes. Utilizing Equation 5 in conjunction with Equation 3, we can derive the following expression:

$$H = GE(\phi(G)) \tag{6}$$

### 3.3 Pre-trainining

We pre-train the model to understand hierarchical structural interactions in CPI tasks, incorporating unimolecular pre-training as necessary. We develop a distance-based self-supervised learning (SSL) task to elucidate the inter and intra relationships between various levels of components in CPI tasks.

Given a complex conformation with spatial coordinates, we deliberately mask the intermolecular distance between the compound and the protein, effectively employing the protein as a reference. We subject the compound to translation and rotation maneuvers, relocating it to an indeterminate position. We strive to comprehend the docking process between the two rigid graphs at distinct hierarchical levels. Consequently, we introduce the SSL loss functions to facilitate this learning process.

Given a graph encoder, we can obtain the graph representation for every node, $H_{v^p}, H_{v^c} = GE(\phi(G))$, and $H_{m^p}, H_{m^l} = GE(\phi(\mathcal{T}))$. We define the predicted distance $S$ as follows:

$$S\left(v^p, v^c\right) = f\left(h_{v^p}, h_{v^c}\right) \tag{7}$$

where $v^p$ represents the node in the protein graph and $v^c$ denotes the node in the compound graph. The function $f(*)$ is the inference mechanism that maps the nodes' representations to the distance between the two corresponding nodes.

Our model aims to capture the interactions at both atomic and motif levels. To achieve this, we employ two separate encoders for these levels. However, since atoms and motifs are interrelated and cannot be considered in isolation. For example, the motif should be a restriction of the atom on the motif. Thus, we introduce a third encoder. This encoder treats the motif as prior knowledge for the atom and subsequently encodes the atom accordingly. During the pre-training stage, we incorporate three interaction losses associated with the three encoders: an atomic distance loss, a motif distance loss, and an atomic distance loss conditioned on motif distance. This approach allows our model to effectively learn the complex interplay between atoms and motifs and their contributions to the overall compound-protein interactions. We define the loss as

$$L = L_V + L_M + L_{V|M} \tag{8}$$

where

$$L_V = \sum_{i=1}^{m} \sum_{j=1}^{n} (Dis(v_i^c, v_j^p) - S(v_i^c, v_j^p))^2 \tag{9}$$

denotes the atomic level intra-loss. We mask the distance between compound nodes and protein nodes on the atom graph. The SSL task is to predict the masked distance.

$$L_M = \sum_{i=1}^{m} \sum_{j=1}^{n} (Dis(m_i^c, m_j^p) - S(m_i^c, m_j^p))^2 \tag{10}$$

denotes the motif level intra-loss. We mask the distance between compound nodes and protein nodes on the motif graph. The SSL task is to predict the masked distance.

$$L_{V|M} = \sum_{i=1}^{m} \sum_{j=1}^{n} (Dis(v_i^c, v_j^p) - S(v_i^c, v_j^p \mid S(M^c, M^p)))^2 \tag{11}$$

represents the inter-loss between the atomic level and motif level. We mask the distance between compound nodes and protein nodes on the atom graph; however, we retain the motif distance as a priori knowledge, establishing the relationship between different hierarchical levels. The SSL task is to predict the masked distance.

## 3.4 FINE-TUNING

In the experimental section, we assess the performance of our proposed model on a classic CPI-related task, namely binding affinity prediction. Additionally, we conduct a case study to demonstrate that our model exhibits a higher degree of self-consistency with respect to established chemical rules.

### 3.4.1 BINDING AFFINITY PREDICTION

Binding affinity is a measure that quantifies the strength of interaction between a compound and a protein. This task involves predicting the binding affinity as a continuous value, thus making it a regression task. After learning representations with the Encoder, we concatenate them on the three encoders to capture complex interactions between the compound and protein. We can easily predict binding affinity as a continuous value using a simple linear head. We defined the binding affinity loss by mean square error (MSE).

## 4 EXPERIMENT

### 4.1 BINDING AFFINITY PREDICTION

#### 4.1.1 DATASET

In this study, we use the PDBBind 2019 dataset (Wang et al., 2004) for training our model on compound-protein interaction (CPI) related tasks. The PDBbind dataset is a comprehensive collection of experimentally measured binding affinities for protein-ligand complexes, curated from the Protein Data Bank (PDB), which is widely used for benchmarking and evaluating computational methods in drug discovery and design. The PDBBind dataset consists of a general set, refined set, and core set, representing data quality from coarse to fine. Additionally, we use the CASF-2016 benchmark (Su et al., 2018) for evaluating our model's performance. CASF-2016 is the core set of PDBBind 2016 and is commonly used in CPI-related tasks.

For our training data, we manually remove the CASF-2016 data and any nonsensical data (e.g., cases where all protein atoms are more than 6 Å away from all compound atoms) from the PDBBind 2019 dataset. This results in a total of 16,493 compound-protein pairs for training. We then randomly split this dataset into training and validation sets with a 9:1 ratio. For CASF-2016 benchmark. After removing nonsensical data from this dataset, we are left with 275 compound-protein pairs to be used for testing our model's performance.

In the process of predicting the label, we utilize the variable $pKa$, signifying the binding affinity within a protein-compound complex. The assessment of binding affinity serves as a crucial parameter for advancements in drug discovery.

### 4.1.2 BASELINES AND METRICS

We select competitive approaches from drug-target binding affinity (DBTA) related literature for comparison. Given the absence of pre-trained models among these competitors, we also establish a pre-trained baseline by adopting the pocket pre-trained model and compound pre-trained model from Uni-Mol (Zhou et al., 2023). We employ a similar methodology as Phi-former for affinity prediction. In order to mitigate the impact of the model itself and demonstrate the efficacy of our pre-training scheme, we conducted an experiment involving training from scratch. This was done by maintaining identical parameters as those in the pre-trained version of the Phi-former, while abstaining from pre-training, thus allowing for a comparative analysis. The primary objective of utilizing this baseline is to demonstrate the effectiveness of modeling hierarchical interactions. Our evaluation metrics include Pearson's correlation coefficient ($R_p$) and Root-Mean Squared Error (RMSE).

| Method | RMSE($\downarrow$) | $R_p(\uparrow)$ |
|---|---|---|
| SIGN (Li et al., 2021) | 1.316 | 0.797 |
| IGN (Jiang et al., 2021) | 1.291 | 0.811 |
| $\triangle VinaRF_{20}$ (Zhu et al., 2020) | | 0.816 |
| OnionNet (Zheng et al., 2019) | 1.278 | 0.816 |
| Mol-PSI (Jiang et al., 2022) | 1.278 | 0.844 |
| GraphDTA (Nguyen et al., 2021) | 1.562 | 0.697 |
| $K_{Deep}$ (Jiménez et al., 2018) | 1.270 | 0.820 |
| SS-GNN (Zhang et al., 2023) | 1.181 | **0.853** |
| Pre-train baseline | 1.418 | 0.758 |
| Phi-former(no pre-train) | 1.379 | 0.771 |
| Phi-former(ours) | **1.159** | 0.846 |

Table 1: Results on binding affinity prediction task

### 4.2 ANALYSIS

The results in Table 1 show that our model outperforms all other models in terms of RMSE, achieving the lowest value of 1.164. This indicates that our model has the smallest average error in its predictions. In terms of Pearson correlation, Phi-former achieves a value of 0.846, which is slightly lower than the best-performing model, SS-GNN, with an $R_p$ of 0.853. Overall, Phi-former demonstrates strong performance in both metrics, suggesting its effectiveness in the field of computer science English writing for the given task.

In comparing with the pre-trained baseline model, it is evident that pre-training on individual models can facilitate the learning of information to a certain degree. Nevertheless, neglecting the interaction process during the pre-training phase is likely to result in inadequate performance in the subsequent affinity-based tasks. Relative to the non-pretrained version, employing pre-training evidently leads to a considerable improvement in the model's effectiveness both on $R_p$ and RMSE.

### 4.3 FUNCTIONAL GROUP INTERACTION CASE STUDY

To ascertain the self-consistency of our proposed model concerning established chemical principles, we undertake an investigative case study. The objective of this investigation is to determine the capability of our model to accurately capture the $\pi - \pi$ interaction. As shown in figure 5(a), $\pi - \pi$ interaction is a quintessential non-covalent attractive force that occurs between two aromatic rings, exemplified by those present in benzene molecules. Given that our pre-training task involves predicting distances, our model is expected to infer the relative spatial positioning of the two aromatic rings within the residue and the ligand.

As illustrated in Figure 5, an examination of the predicted distances of the pre-training model reveals a notable observation. The model constructed solely on the atomic level, as depicted in Figure 5(c), is unable to accurately capture the $\pi - \pi$ interaction (Figure 5(a) is a classical $\pi - \pi$ interaction case). The distance between the two benzene rings is predicted to be 6 Å, which implies a lack of interaction between them. Conversely, when a motif level model is incorporated as a constraint, as

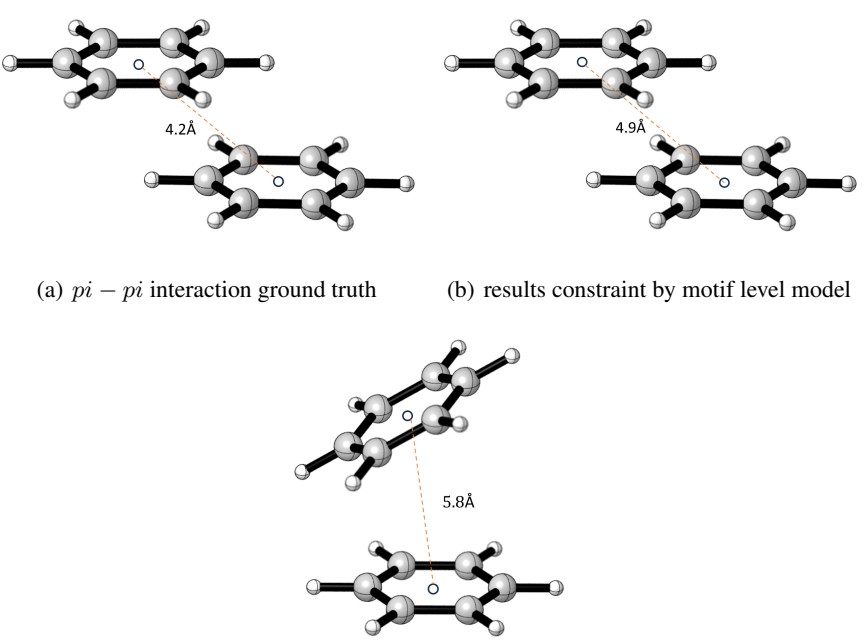

(a) $pi-pi$ interaction ground truth  (b) results constraint by motif level model

(c) results only on atomic level

Figure 5: A case study demonstrates that, under the constraint of a motif, the model can effectively comprehend certain non-covalent interaction principles.

depicted in Figure 5(b), our model successfully deduces the relative spatial positioning of the two aromatic rings within the residue and the ligand. This demonstrates that our model not only exhibits high effectiveness but also adheres to chemical rules in a self-consistent manner. This case study underscores the ability of our model to learn and represent significant non-covalent interactions, such as the $\pi-\pi$ interaction, while maintaining chemical consistency.

## 4.4 CONCLUSION AND FUTURE WORK

In conclusion, this study demonstrates the importance of considering the role of motifs in predicting compound-protein interactions (CPIs) for AI-aided drug design. By proposing a pair-wise hierarchical interaction representation learning (Phi-former) method, we can model molecular interactions more effectively, taking into account atom-atom, motif-motif, and atom-motif interactions. The Hir method employs a pair-wise specific pre-training framework and incorporates two intra-losses and one inter-loss for predicting 3D structure information, allowing for more systematic and mutually beneficial learning of different interaction levels. Our results show that the Phi-former method achieves superior performance on CPI-related tasks, and a case study provides evidence of the method's ability to accurately identify specific atoms or motifs activated in CPIs. This work highlights the potential for further development and application of deep learning methods that consider hierarchical structures in drug design and other related fields.

In future research endeavors, we plan to enhance the Phi-former model by incorporating both pre-training and fine-tuning stages, with the objective of training a more comprehensive model on an extensive dataset. This will enable the model to excel not only in the realm of compound-protein interaction (CPI) tasks but also in the domains of drug-drug interaction (DDI) and protein-protein interaction (PPI) tasks.

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
