# A  APPENDIX

## A.1  MODEL INITIAL INPUT

Given an input graph $G(V, E, P)$, our objective is to determine the input node feature matrix $X$ and spatial positional encoding $S$ utilizing Equation 3. In this section, we present the methodology for obtaining $X$. For the atomic-level approach, we initially obtain the node embedding from the pre-trained Uni-Mol model Zhou et al. (2023). Since Uni-Mol solely incorporates atom type information, we manually augment the node features with additional attributes and concatenate them accordingly.

| number | feature | range |
|--------|---------|-------|
| 1 | chirality | [0, 3] |
| 2 | charge | [0, 10] |
| 3 | hyb | [0, 6] |
| 4 | numH | [0, 8] |
| 5 | valence | [0, 7] |
| 6 | degree | [0, 10] |
| 7 | aromatic | [0, 1] |
| 8 | is_protein | [0, 1] |

Table 2: Node feature table