# OpenReview forum: "Phi-Former: A Pairwise Hierarchical Approach for Compound-Protein Interaction Prediction"
_ICLR.cc/2024/Conference — ICLR 2024 Conference Withdrawn Submission_

### Official Review · Reviewer_1zG2 · 2023-10-29

**Soundness:** 2 fair
**Presentation:** 1 poor
**Contribution:** 2 fair
**Rating:** 1
**Confidence:** 4

**Summary:**

Considering the functions of molecular fragments (i.e.,motifs) that participate in the interactions of moleculars, this paper proposes Phi-former, a pair-wise hierarchical interaction representation learning model, which incorporates a pair-wise specific pre-training framework to effectively capture and simulate interactions at several levels, including atom-atom, motif-motif, and atom-motif interactions. Phi-former exhibits enhanced performance in tasks related to CPI.

**Strengths:**

This paper proposed Phi-former, which is pre-training framework to effectively capture and simulate interactions at several levels, including atom-atom, motif-motif, and atom-motif interactions to model the interactions between compounds and proteins.  The inter and intra losses on pre-training tasks, are designed to help the model to learn across different interaction levels. Case studies justified the proposed model.

**Weaknesses:**

1. In Figure 1 (C),  without these constraints, an erroneous binding result of the weak interaction is presented. What are the constraints? Does it only mean the motifs?
2. The inter- and intra-losses enable the learning of different interaction levels, which may include $L_V, L_M, L_{V|M}$; however, there are no abations about these inter- and intra-losses to present their effects.
3. In Eq. (1), what do Q and q_1, q_2,...,q_k mean? If n denotes the number of atoms, what about k? I do advise the author to check all the formulas that appear in the paper. E.g., in Eq.(2), how to calculate d_{ij}, what is S in Eq. (3), what are the relationships between s_{ij}^{k} and S?
In Eq.(5), the authors define the graph transformer as GE. Why can you give such definitions? Where are the learnable parameters of the graph transformer? If you have seen similar definitions, please give citations. Worse, you have given the equations (3) and (5), where are the necessities of Eq.(6)?  For Figure 5, it doesn't need to take up so much space.
4. There is only one downstream task to evaluate your proposed pre-training model, Phi-former, and only 16,493 compound-protein pairs for training. Could you evaluate Phi-former on more tasks? Besides, in Table 1, point estimation is often random and inaccurate; could you give std values?
5. The most incredible thing is in Section 4.2 Analysis, the authors say Phi-former demonstrates strong performance in both metrics, suggesting its effectiveness in the field of computer science English writing for the given task. What does this mean? Phi-former is proposed for COMPOUND-PROTEIN INTERACTIONS PREDICTION, and Table is the task related to binding affinity prediction. Why is it effective in the field of computer science English writing?
6. From the descriptions in Appendix A.1, the node feature matrix X is obtained from the node embedding from the pre-trained Uni-Mol model [1]. X  is from a pre-trained model, and Phi-former is more like a fine-tuning model. Which dataset does the Phi-former pre-train? Why say it is a pre-training model? Incredibly, in Table 1, there are no comparisons with Uni-Mol? I assume the effects are from Uni-Mol.


[1] Gengmo Zhou, Zhifeng Gao, Qiankun Ding, Hang Zheng, Hongteng Xu, Zhewei Wei, Linfeng Zhang, and Guolin Ke. Uni-mol: A universal 3d molecular representation learning framework. 2023.

Typos:

(1) Hierarchical modeling on biomoleculars The models under discussion predominantly focus on the representation of atomic level graphs. This sentence may lack a dot.

(2) graph vertices as shown in Figure 3.1, It can be observed that the single torsional bond is dissociated. Capital and small letter.

(3) Graph Neural Networks (GNNs) repeat in Section 2 Related Works and Section 3.2 Graph Transformer. Whether to use abbreviations.

(4) Table 1 lacks a bottom line.

I sincerely recommend the authors polish their presentations, experiments, etc.

**Questions:**

See questions in the Weakness.

---

### Official Review · Reviewer_M7EC · 2023-10-31

**Soundness:** 3 good
**Presentation:** 2 fair
**Contribution:** 2 fair
**Rating:** 3
**Confidence:** 4

**Summary:**

The paper proposes Phi-former, a pairwise hierarchical interaction representation learning framework for predicting compound-protein interactions (CPIs). The method represents compounds and proteins hierarchically and employs a pairwise specific pre-training framework to model interactions at different levels, including atom-atom, motif-motif, and atom-motif. The authors introduce an intra-level and inter-level Phi-former pipeline for learning the pairwise biomolecular graph representation, which makes learning different interaction levels mutually beneficial. The paper demonstrates that Phi-former achieves superior performance on CPI-related tasks and can accurately identify specific atoms or motifs activated in CPIs, providing good model explanations.

**Strengths:**

1. The visualization plots make this paper easy to understand.
2. The motivation is clear and intuitive.
3. Represents compounds and proteins hierarchically for more systematic modeling of interactions is reasonable.

**Weaknesses:**

1. The method may be computationally intensive due to the hierarchical representation and pairwise specific pre-training framework. The authors should provide some comparison of computational complexity.
2. This paper could provide more details on the implementation of the method.
3. This paper could include more comparisons with existing methods to better highlight the advantages of the proposed method.

**Questions:**

1. How does the computational cost of the proposed method compare to existing methods?
2. What does "English writing" mean on page 8?
3. How does the method handle large-scale datasets?
4. Can the method be extended to predict other properties of compound-protein interactions, such as binding affinity?

---

### Official Review · Reviewer_smio · 2023-10-31

**Soundness:** 3 good
**Presentation:** 3 good
**Contribution:** 2 fair
**Rating:** 5
**Confidence:** 5

**Summary:**

The paper addresses the challenge of predicting compound-protein interactions (CPIs) crucial for AI-aided drug design. It critiques the limitations of current deep learning methods in modeling molecular interactions, which don't fully align with the chemical realities of CPIs, particularly the dominant role of molecular fragments or motifs. The proposed solution, a pair-wise hierarchical interaction representation learning method named Phi-former, aims to better capture these interactions by considering the role of motifs. This method utilizes a hierarchical representation of compounds and proteins, emphasizing a systematic approach to modeling interactions at multiple levels - atom-atom, motif-motif, and atom-motif. The authors claim that Phi-former achieves superior performance in CPI-related tasks and can provide insightful model explanations beneficial for molecular structural optimization.

**Strengths:**

- The concept of utilizing a hierarchical interaction representation and focusing on motifs in CPIs is a notable advancement in the field of AI-driven drug design. The Phi-former approach appears to introduce a novel perspective in this domain, potentially filling a significant gap left by traditional energy-based and recent deep learning methods.
- The case study indicating Phi-former's ability to identify specific atoms or motifs involved in CPIs is promising, particularly from the standpoint of model interpretability and applications in molecular structural optimization. This aspect sets the paper apart, as it not only focuses on predictive accuracy but also on the explainability of the AI model, which is crucial for practical applications in drug design.

**Weaknesses:**

- Experimental Validation
  - The claim of superior performance by Phi-former in CPI-related tasks is impressive. Yet, the paper only shows a comparison on one dataset, it would be more convincing if provided more quantitative data and comparisons with existing methods to substantiate this claim. A broader statistical analysis and error metrics could enhance the credibility and allow for a better understanding of its practical efficacy.
- Methodological Details and Validation
  - Specifics of Graph Transformer: The intricacies of the graph transformer, particularly how it handles the complexities of 3D structures in biomolecules, might require more detailed explanation or validation.
  - Encoder Coordination: How the three encoders' outputs are combined and whether this integration might lead to loss or dilution of specific molecular features aren't thoroughly explained.
- Technical Assumptions and Limitations
  - Assumptions in Spatial Positional Encoding: The effectiveness and assumptions behind the spatial positional encoding method, especially in terms of rotation or translation invariance, might require more scrutiny.
  - Data Requirements: The model's performance relative to the quality and quantity of training data hasn't been addressed. It's unclear how sensitive the model is to variations in data quality.

**Questions:**

- Can you elaborate on the computational resources required for your model, particularly in terms of time and hardware, for both training and inference phases?
- How did you determine the proportion and pattern of manual masking in the pre-training phase? Could you provide more insight into how this strategy reflects realistic molecular interactions?
- Could you detail how the graph transformer architecture effectively captures and utilizes the 3D spatial information of biomolecules, particularly in complex, flexible structures?
- How are the outputs of the three different encoders integrated? Is there a risk of information loss or dilution, and if so, how is this mitigated?

---

### Official Review · Reviewer_b38p · 2023-11-01

**Soundness:** 2 fair
**Presentation:** 2 fair
**Contribution:** 1 poor
**Rating:** 3
**Confidence:** 4

**Summary:**

This paper proposes a new method for predicting compound-protein interactions (CPIs) using a pair-wise hierarchical approach that considers both atomic and motif levels of molecular structures. The method, called Phi-former, uses graph transformers to encode the compound and protein graphs and employs a pre-training and fine-tuning scheme to learn the interactions in different levels. The paper demonstrates that Phi-former can achieve superior performance on binding affinity prediction. The paper claims that Phi-former is more effective and consistent with chemical principles than existing methods that ignore the role of motifs.

**Strengths:**

S1. The paper studies a significant problem in drug discovery.


S2. The paper is with sufficient motivation to consider motif-level information.


S3. The paper employs a self-supervised task to incorporate spatial distance information.

**Weaknesses:**

W1. The novelty of the proposed method is not clear enough. The graph learning architecture is quite similar to the existing molecular attention transformer [1], which also use distance-based attention bias. Moreover, although authors claim they proposed distance-based self-supervised learning in CPI tasks, applying the distance prediction loss has been proposed in previous works for molecular structure learning [2]. The authors need to highlight their key contributions and differentiate their method from previous works.


W2. The experiments are insufficient and not convincing.

(w2.1)	Firstly, the authors claim they "manually remove any nonsensical data (e.g., cases where all protein atoms are more than 6 A away from all compound atoms)." Why such data is nonsensical?


(w2.2)	Secondly, for baseline comparison, it seems that the authors just simply use the reported experimental results from the original papers of some baseline models. It is quite not reasonable. Since the authors removed several "nonsensical data" from the CASF-2016 benchmark, the testing set is not fully the same as these baseline models. On the other hand, the training set is also different. For instance, the result of RMSE (1.316) is derived from the SIGN model training on the refined set (less than 4000 samples), but the model of this paper is trained on 16493 samples.



W3. The framework description is not clear. Some method details are missing.

(w3.1)	For instance, in Section 3.1, it is not clear what does "qn" mean, and "q_1 = ave{q1, q2, …, qk}" is also ambiguous. How to construct the motif-level graph is not introduced in detail. More descriptions are necessary to show the process of graph construction.


(w3.2)	In section 3.3, what does "S(v^c_i, v_j^p | S(M^c, M^p))" mean? How to implement such operation in practice?



W4. It would be better to evaluate the computational cost of the new method, as the efficiency is a key factor in drug discovery applications with large-scale and complex molecular structures.



[1] Maziarka, Łukasz, et al. "Molecule-augmented attention transformer." Workshop on Graph Representation Learning, Neural Information Processing Systems. 2019.


[2] Fang, Xiaomin, et al. "Geometry-enhanced molecular representation learning for property prediction." Nature Machine Intelligence 4.2 (2022): 127-134.

**Questions:**

Q1) Please refer to W2 and specify the question about the experiment settings.

Q2) Please refer to W3 and clarify the presentation.

Q3) In the introduction, the authors said “We design three representative CPI tasks in the fine-tuning phase: binding affinity prediction (Wan &Zeng, 2016), drug-target interaction (DTI) (Bleakley & Yamanishi, 2009), and docking pose generation. Our pre-trained model demonstrates promising performance across these tasks.“ But in the experiment part, why there is only one fine-tuning task about Binding affinity prediction?